# Recent Advances in the Microencapsulation of Essential Oils, Lipids, and Compound Lipids through Spray Drying: A Review

**DOI:** 10.3390/pharmaceutics15051490

**Published:** 2023-05-13

**Authors:** Diego Mauricio Sánchez-Osorno, María Camila López-Jaramillo, Angie Vanesa Caicedo Paz, Aída Luz Villa, María S. Peresin, Julián Paul Martínez-Galán

**Affiliations:** 1Grupo de Investigación Alimentación y Nutrición Humana-GIANH, Escuela de Nutrición y Dietética, Universidad de Antioquia, Cl. 67, No 53-108, Medellín 050010, Colombia; 2Grupo de Investigación e Innovación Ambiental (GIIAM), Institución Universitaria Pascual Bravo, Cl. 73, No 73a-226, Medellín 050034, Colombia; 3Grupo Catálisis Ambiental, Universidad de Antioquia, Cl. 67, No 53-108, Medellín 050010, Colombia; 4Sustainable Bio-Based Materials Lab, Forest Products Development Center, College of Forestry, Wildlife, Auburn University, Auburn, AL 36849, USA

**Keywords:** spray drying, essential oils, lipids, compound lipids, encapsulation, structured lipids

## Abstract

In recent decades, the microcapsules of lipids, compound lipids, and essential oils, have found numerous potential practical applications in food, textiles, agricultural products, as well as pharmaceuticals. This article discusses the encapsulation of fat-soluble vitamins, essential oils, polyunsaturated fatty acids, and structured lipids. Consequently, the compiled information establishes the criteria to better select encapsulating agents as well as combinations of encapsulating agents best suited to the types of active ingredient to be encapsulated. This review shows a trend towards applications in food and pharmacology as well as the increase in research related to microencapsulation by the spray drying of vitamins A and E, as well as fish oil, thanks to its contribution of omega 3 and omega 6. There is also an increase in articles in which spray drying is combined with other encapsulation techniques, or modifications to the conventional spray drying system.

## 1. Introduction

Bioactive ingredients have become an important component in the diet of humans, particularly because consumers are more aware of the direct impact of food on human health [1]. Some statistics show that consumers are more keen to consume functional foods, as reported by Market Research Transparency in 2016. It is expected that the functional food global market could exceed more than 309 billion on sale by 2027 [2]. According to technological advances, functional food products have gained special attention on the design and manufacture of both functional and nutraceuticals food types through the incorporation of bioactive ingredients [3]. Functional ingredients are of paramount importance for obtaining stable microparticles that maintain bioactive ingredients aimed at providing protection against different physical or chemical reactions that could potentially conduce to degradation [4]. The production of functional foods implies some challenges, both in the incorporation of the bioactive ingredients in the food matrix, demonstrating their stability during the processing period, but also their stability during storage. Additionally, interactions should be certainly minimized. It is also essential to guarantee the continued functionality of the bioactive ingredients until they reach the gastro-intestinal tract, to be released at the targeted site [5]. The above microencapsulation requirements seek to ensure both protection and release under target conditions; however, these are not the only parameters that must be considered during the encapsulation of bioactive ingredients [6]. In addition, it is important to understand the characteristics of bioactive compounds such as solubility, chemical structure, and physiological functions, which allows to determine protection requirements against various environmental factors. Knowing that lipids are fats and fat-like substances, which differ in chemical composition, structure, and by physiological and biochemical function [7], it could be possible to predict the optimum encapsulation process, as well as encapsulation agents.

Lipids, compound lipids and essential oils are particularly sensitive to heat and susceptible to factors such as oxidation, humidity, light, and air [6,8], as such conditions limit their application. Microencapsulation is widely used to extend the stability of bioactive lipidic compounds through processing, storage, and handling [5]. In this sense, the structured lipids have gained commercial relevance as it is possible to optimize the bioavailability of individual or the totality of the component fatty acids within the triacylglyceride or phospholipid in question. A structured lipid, e.g., a structured triacylglyceride, is a “tailor made” molecule formulated for a specific nutritional and/or technological function. Structured lipids have gained relevance as it is possible to optimize the bioavailability of the individual or the totality of the component fatty acids within the triacylglyceride or phospholipid in question. This would allow to determine the type of fatty acids, as well as their order for the desired structure [9].

Microencapsulation technology is as a process for coating bioactive ingredients in the form of droplets. Such bioactive ingredients could be incorporated evenly in a homogeneous or heterogeneous matrix, obtaining as a final result small capsules that allow the protection of the active ingredient within the core, while allowing their controlled release [10]. The microencapsulation process encompasses a set of steps or conditions required to protect the bioactive ingredients of solid, liquid, and gaseous materials observed as micro or nano capsules, with the characteristic of releasing their contents under conditions and quantities specifics through the time [11]. Among the areas in which microencapsulation technology has been used are biomedical, textiles, and medicine, and it has also been widely used in pharmaceutical applications, as well as food industries [12]. The main reasons why microencapsulation is of such relevance in food and pharma are associated with: (i) reducing the interaction of the bioactive ingredients in different environments (heat, humidity, air and light); (ii) reducing the probability of losing bioactivity; (iii) facilitating handling; (iv) releasing under specific conditions the encapsulated, active ingredient; (v) preventing the occurrence of undesired flavors and odors; and (vi) the dosing of the bioactive ingredient when used in small quantities [13].

As previously mentioned, one of the most common uses of microencapsulation technology is the production of high value-added food with the aim of obtaining stable functional foods that possess desirable, specific properties, such as mentioned before. The cost is one of the most prevalent variables that will determine the applicability of the microencapsulation process for a certain product. Low microcapsule production cost will allow for maintaining a competitive price without compromising product quality, especially when it comes to health-related products [11]. At the industrial/commercial level, there are three aspects that must be evaluated during the micro encapsulation process, these are: (a) the production cost of the microcapsules; (b) the possibility that said microcapsules can be used in various products, or in a wide range of products; and (c) the possibility of scaling-up from proof of concept to large-scale production.

Despite the widespread use of the spray drying technique for the production of microcapsules, it is important to highlight other encapsulation techniques that have gained strength in recent years. Among these are coacervation and molecular inclusion, as well as fluidized bed, and coacervation. All microencapsulation techniques are governed by the same basic principle, which is the protection of a specific compound. However, the technique of choice is governed by aspects such as the properties of the bioactive ingredient (physical and chemical), desired particle size, the physical-chemical properties of the encapsulating material or substrate, production costs associated with microencapsulation, and controlled release mechanisms [14,15]. In light of the aforementioned factors, spray drying has widely demonstrated attractive advantages over other encapsulation techniques, such as: (1) low microcapsule manufacturing costs; (2) the wide variety of available encapsulating agents; (3) the high retention and stability of volatile compounds in the final product; and (4) the possibility of scaling at an industrial level allowing for large-scale production [16,17,18,19].

This review aims to ask the question: are there significant advances in the encapsulation of essential oils, lipids, or compound lipids, by spray drying? This review considers the main encapsulating agents and bioactive compounds used in the spray drying of essential oils and polyunsaturated fatty acids. It includes a theoretical framework of the encapsulation process, the most representative encapsulation techniques, and also describes the spray drying encapsulation methodology. Additionally, the spray drying microencapsulation method will be discussed, followed by an overview of relevant literature published in the space of the microencapsulation of different lipids, and its comparison with other microencapsulation techniques. 

## 2. Materials and Methods

### 2.1. Methods

The period between the years 2000 and 2021 was selected in order to know the progress of the subject of the encapsulation of essential oils, lipids, and lipid components, by spray drying during the course of the 21st century. Additionally, it seeks to identify the main encapsulating materials and the appearance of new encapsulating materials over time.

Phase 1: Information sources and keywords were identified in Scopus (“encapsulation”, “spray dry”, “essential oil”, “lipids”, and “compound lipid”). The search criteria was selected, focusing on the years 2000–2021. The search equation was TITLE-ABS-KEY ((“essential oil” OR “lipids” OR “lipid compound”) AND encapsulation AND “spray drying”), which led to a total of 382 articles (Figure 1).

Phase 2: A table format was created that contains the following criteria: article name, author, year, country, keywords, abstract, journal, Scimago quartile, and explicit mention of spray drying and encapsulation of essential oil or lipids or compound lipids.

Phase 3: The analysis was carried out in two thematic axes found in 120 papers: axis 1 related to encapsulation by spray drying and axis 2 on the encapsulation of essential oils, or lipids, or compound lipids.

### 2.2. Theortical Framework of the Microencapsulation

In microencapsulation, small particles of an active ingredient are coated or trapped within a film composed of one or more materials [20,21,22]. The active ingredient to be protected is known as the core material, bioactive ingredient, or internal phase, and can be encapsulated pure or in combination with other materials [12,17,23]. On the other hand, the coating material is called encapsulating agent or wall material [14,17]. The coating can be composed by a mixture of materials that have different physical and chemical properties, thus allowing to obtain a reinforced coating adapted to specific conditions [10,24].

The coacervation technique for microencapsulation has been recorded to be applied for the first time at an industrial level in 1950, when the National Cash Register Company developed “carbonless carbon paper” [25], and it has been used ever since then. Currently, microencapsulation is used in various productive sectors such as textiles [26,27,28], agricultural [10,20,29], cosmetics [30,31,32,33], and pharmaceuticals [13,16,28,29,34]. In the food industry, microencapsulation is utilized to avoid the loss of flavor during processing or storage, because they are usually volatile compounds, highly sensitive to environmental conditions such as light, heat, humidity, oxygen, and radiation, among others, thus reducing unwanted interactions of the bioactive ingredients with the exterior environment [31,35]. Other advantages associated with microencapsulation include the conservation of nutritional value, controlling the release of the encapsulation, the ability to mask some undesired flavors of certain compounds, and facilitating the handling and transportation of materials by changing their original shape and volume [11,20,36].

The size, shape, and structure of these microcapsules is a function of the selected type of encapsulating material, the active ingredient to be encapsulated, and microencapsulation technique [14,31,37]. On the other hand, encapsulation efficiency, the stability of the microparticles, and the characteristics of the final product, are directly related to the encapsulating material of choice. The final particles may present different structures and morphologies, such as single-core, multinuclear, nonspherical multinuclear, multilayer, or multicore (Figure 2).

In order to predict the physical and chemical interactions that could occur between the bioactive ingredient and wall material, it is necessary to know the encapsulating material and its properties. Ideally, the encapsulating agent will present the following characteristics: (1) must not react with the encapsulated active ingredient; (2) must protect the encapsulated ingredient from the outside environment; (3) it must not be toxic; and (4) should not be cost prohibitive, among others [38,39]. The techniques used for the encapsulation of compounds are varied. The selection of the encapsulation technique to be used is carried out considering the physical and chemical properties of both the encapsulating agent and the encapsulated agent. Additionally, by knowing the targeted particle size as well as its shape, release rate, process, and its possible scaling, it is of paramount importance for the selection of the correct microencapsulation technique [40].

#### 2.2.1. Representative Encapsulation Techniques 

Currently, there are various encapsulation methods used at both laboratory and industrial level. Over time, and with increasing technological developments, new encapsulation techniques will continue to emerge and adapt to the needs for various bioactive compounds to be protected, encapsulating substrate materials and market needs. Microencapsulation was originally used to protect the active ingredients against environmental and/or external factors [41,42]; however, over time, controlled release has become a fundamental factor in the development of encapsulation systems [43,44]. Encapsulation techniques can be classified into chemical, physical-chemical, or mechanical [41,45] techniques. Among the most important techniques used for lipids protection are: coacervation; spray drying; liposomes; fluid bed spray coating; and emulsification solvent evaporation/extraction (Figure 3). These encapsulation techniques are explained below.

-Spray drying is a widely used manufacturing technique used in food, agri-food, and the pharmaceutical industry, among others. This technology allows to obtain a powder product, starting from a concentrated liquid solution or suspension. Its operating principle is based on the atomization of the solution, thus generating small micro droplets upon contact with a sprayed stream of hot air (between 150 °C to 300 °C) [45].-In the case of coacervation, a homogeneous solution of charged macromolecules separates into two liquid phases in equilibrium, a colloidal suspension in which the more concentrated phase is known as the “coacervated phase”, and the other is known as the “equilibrium phase”. During the simple coacervation process of a polyelectrolyte, the addition of salt or alcohol normally promotes phase segregation, through the self-neutralization of the loads [45,46]. Additionally, the coacervation complex involves the interaction of at least two biopolymers fillers. Parameters such as concentration, biopolymers ratio, temperature, pH, ionic strength, and charge density, must be balanced for an efficient phase separation [47,48].-Liposomes are vesicular structures made up of lipids organized in bilayers or multilayers. These structures are similar to the structure of the membrane lipid. In the medical field, model studies have been developed of the physical behavior and chemistry of the cell membrane, cell compartments, and the cells themselves vesicular transport structures in and out of the cell. Liposomes are considered vesicles with a unilamellar spherical structure or multilamellar; that is, its vesicular lipid conformation may have a bilayer lipid or several concentric lipid bilayers. By their conformation, unilamellar liposomes are classified into small (SUV), medium (MUV), large (LUV), or giant (GUV) [49,50].-Spray coating in a fluidized bed system allows for obtaining a suitable surface coating by uniformly applying the coating material. In this method, the coating material, which is in liquid state, is sprayed onto all the particles as they move in the fluidized bed [51,52], at the time that the aqueous or organic solution is evaporated, forming the coating layer. With this technique, it is possible to obtain particle size of 100 μm up to 3 mm [52,53]. On the other hand, emulsification solvent evaporation/extraction consists in forming an emulsion by combining a polymer and a volatile organic solvent. By heating the emulsion, the solvent is evaporated [54,55]. The solution is formed by the dispersion of the active component and the encapsulating agent, this dispersion is emulsified in an external aqueous phase, in which the polymer is insoluble. In this technique, the use of stabilizing agents that favor the formation of the particles and the retention of the bioactive compound inside each capsule is common [55]. This technique is normally used to microencapsulate hydrophobic bioactive compounds; it can additionally perform both a single emulsion and a double emulsion, also known as water-in-oil-in-water (w/o/w) [56].

#### 2.2.2. Spray Drying Encapsulation

The spray drying technique has been used to encapsulate bioactive compounds in food ingredients. This technique consists of evaporating the solvent (generally water) rapidly, generating a capsule in which the compound of interest is almost instantly trapped [57,58]. The material used for encapsulation by this method, as well as the operating conditions, are of vital importance to achieve a correct microencapsulation and the adequate protection of the active principle. In spray drying, a liquid mixture is sprayed into a stream of a gas; normally air, although an inert gas, such as nitrogen, can also be used (Figure 4).

In this method, the liquid to be dried (encapsulating an active ingredient to be encapsulated) is flowed to a nozzle where it is atomized into the drying chamber, in such a way that the liquid increases the surface area in contact with hot air, which allows its complete drying in a short period of time. In this way, the sensible heat of the supplied hot air is used for the evaporation of the solvent, turning the droplets into a fine powder in less than a second [57,59]. In general, the spray drying process reduces the water content and the aqueous activity in the dehydrated material, helping the microbiological stability of the food, preventing chemical or biological degradation, reducing storage costs, and also obtaining a product with specific and desired properties [60,61].

This process is considered the most common and economical technique for producing microencapsulated food materials, for example, when compared with lyophilization. Spray drying is between 30 and 50 times cheaper [45,62], even knowing that this technique is considered low energy efficiency, since it impossible to use all the energy available as heat in the drying chamber [63,64]. The main encapsulating materials that have been reported include acacia gum, maltodextrins, modified starch, and their mixtures [65]. To a lesser extent, polysaccharides (alginate, carboxymethylcellulose, guar gum) and proteins (serum milk, soybeans, sodium caseinate) have also been used. In these cases, costs are higher due to their low solubility and the need to evaporate a greater quantity of liquid to produce the same quantity of dry matter [19,65]. The main advantages of spray drying include high level of homogeneity in production, continuous and controlled process, as well as ease to operate and high performance.

## 3. Results

After the de-analysis of the selected articles, considering the general aspects of each paper and carrying out a classification related to aspects such as country, year, and number of publications, some graphics were built. Additionally, the types of wall materials employed in spray drying microencapsulation, applications, and journals, were separated aiming to visualize all the information in an easier way. Finally, patent analysis related to spray drying encapsulation of essential oils, lipids, or compound lipids, were elaborated aiming to show some important industrial and academy discoveries, or novelties.

### 3.1. Relevance of Technology per Geographical Regions and Market Push 

Figure 5 shows countries’ publications with respect to the most common encapsulation techniques used to protect essential oils, lipids, or compound lipids. The highest number of publications corresponds to Brazil (16 publications). The Latin American food encapsulation market is driven by factors such as the growing popularity of exotic flavors, cuisines and gourmet varieties, changing dietary habits, rising importance of the preservation of food products, and the growing speed and agility in new product innovation [66]. 

Additionally, the Latin American food encapsulation market was worth USD 4.46 billion in 2021 and estimated to reach USD 6.09 billion by the end of 2026, with a growing potential of 6.43%, where Brazil, Argentina, and Chile are the main contributors [66]. Worldwide, 10 essential oils represent 85% of the market: orange, lemon, mint, java-citronella, cedar, eucalyptus, spice with citral such as may chang and lemongrass, lavender, lavandins, and pines. Brazil is located as the fourth largest exporter of oils essential [67]. 

In the year 2020, Brazil has been reported as the country with the greatest export potential in Latin America of a great variety of products, among which are: orange essential oils; lemon essential oils; citrus essential oils; and essential oils concentrated in fats and fixed oils [68]. The food encapsulation market is estimated to be valued at USD 9.9 billion in 2020 and is projected to grow at a CAGR (Compound Annual Growth Rate) of 7.5%, recording a value of USD 14.1 billion by 2025 [69].

India and China are the next countries with the most active research in this area. In the case of both countries the highest number of publication (12 and 10 publications, respectively,) could be due to the fact that they are the most populated countries in the world with 18% and 19%, respectively [70]. Additionally, in both India and China there has been a millenary tradition in the production of essential oils. These two countries are among the main exporters of essential oils worldwide. At 24.20 million tons (mt) in 2016–17, and an estimated 23.95 mt in 2017–2018, India’s edible oil consumption ranks number two globally, behind China (35 mt). Seventy percent (14 mt) of this demand is satisfied by imports, composed mainly of palm oil (9.5 mt), soybean (2.99 mt), and sunflower oil (1.54 mt) [71].

Additionally, initiatives to develop collaborations between universities from different countries and institutions have been developed, such as Centro de Educação Superior da Região Sul (Brazil), Centro Tecnológico de la Carne de Galicia (Spain), Nutrition and Food Science Area, Preventive Medicine and Public Health, Food Sciences, Toxicology and Forensic Medicine Department (Spain), Universidade Federal de Santa Maria (Brazil), and the Department of Animal Source Food Technology (Serbia); as well as the increasing partnership between universities or private sector from the European Union with Latin America countries and United States.

### 3.2. Years and Publications

Figure 6 shows the trends of spray drying encapsulations for essential oils, lipids, or compound lipid publications, according to a year in the last 20 years. The figure is evidence of a growing behavior of spray drying encapsulations for essential oils, lipids, or compound lipids along time between 2000 and 2021. The number of publications related to “microencapsulation” (February 2021), reaches 13,979 and, of these, 382 are simultaneously related to “encapsulation”, “spray drying”, “essential oil”, “lipids”, or “compound lipid”. From 2000 to 2010 there were only 49 papers published, while between 2011 and 2020, 333 papers were generated (source: www.scopus.com, 18 February 2021). The rise of publication numbers from 2010 could be due to the application of new wall materials, as well as new ingredients with possible health benefits [72]. 

The years 2015 and 2018 showed the greatest growth in the number of patents, with China leading the domains of origin, inventors, and owners of microencapsulation technology. The largest number of applications of microcapsules were observed in the food industry, and the foods containing microencapsulated oils were powdered seasonings, dairy products, rice flour, nutritional formulae, pasta, nutritional supplements, and bread. The increase in the oxidative stabilities of oils was the most cited objective to microencapsulate oils [73].

Furthermore, during the year of the COVID-19 pandemic (2020), the number of publications doubled, compared to 2019.

### 3.3. Journals

Microencapsulation has been utilized to enhance the discharge properties of medications, to cover unpleasant taste, and to give specific tranquilize conveyance in pharmaceutical enterprises [74]. Additionally, it has been used in various industries, including the make-up, pharmaceutical, agrochemical, and food business, among others. 

The increasing interest in spray drying encapsulations for essential oils, lipids, or compound lipids, and therefore creating a large number of indexed journals, is shown in Figure 7 and Figure 8, and are related to economic benefits and increasing research in these fields [75]. Figure 7 and Figure 8 evidence some journals related to the food science of pharmacology, which have published papers associated with spray drying encapsulation and essential oils, lipids, or compound lipids. The total number of journals was divided into two groups, the journals with a Q1 classification (Figure 7) and the journals with a Q2, Q3, and Q4 classification (Figure 8). 

#### 3.3.1. Q1 Articles 

From this group, the journal “Trends in Food Science and Technology “is the one with the most publications related to this review topic (9%) from the United Kingdom, associated with food science and biotechnology with a Q1 quartile [76]. 

Additionally, the second journal with the most publications is “Food Research International” (6%) from the United Kingdom, with the food science topic a Q1 quartile [77]. 

Other journals, such as “Macromolecular Symposia” (5%), “Journal of Food Engineering” (5%), and “Food Chemistry” (5%), also evidence that the topic of the present paper is the subject of wide research.

#### 3.3.2. Q2, Q3, and Q4 Articles 

Among this second group, the percentage of participation from each journal has more uniformity. The first journal with more publications around this topic is “International Journal of Cosmetic Science” (9.9%) from the United Kingdom, with chemistry, dermatology, and pharmaceutical science, with a Q2 quartile and aging, colloid and surface chemistry, and drug discovery, with a Q3 quartile [78]. 

The trend of these papers are related to characteristics such as spray drying as the main encapsulation technique, main wall materials employed and industrial applications. Hence, the largest distribution of publications is not grouped into a single journal, but 65% of the journals are in the Q1 quartile, and 35% of the journal of this research are Q2, Q3, and Q4, indicating the quality and validity of research found during this review and the numerous countries, research groups, and industrial interest focusing on new wall material and the improved protection of essential oils, lipids, and compound lipids [79].

## 4. Discussion

### 4.1. Substrates Used in Microencapsulation

There are a huge variety of materials used for microencapsulation; however, all of them must meet certain basic characteristics such as: promote release in accordance with the objectives set for the bioactive compound; protect the bioactive compound from environmental factors such as light, heat, and oxygen, among others; do not react with the bioactive compound; decrease the diffusion of the bioactive compound to the external environment; be chemically compatible; be easy to handle; and be waterproof, stable, and flexible [11,22,28,80,81]. Encapsulated materials can be natural, semi-synthetic, or synthetic polymers [19,82]. Among the most used encapsulating agents for lipids, lipophilic compounds, or essential oils are gum arabic, barley protein, carrageenan, chitosan, maltodextrin, methylcellulose, skim milk powder, and whey protein (Table 1). Additionally, Table 1 shows some common encapsulating agents used in microencapsulation processes, as well as some of the common combinations among the different encapsulating agents used for protecting lipid components.

#### 4.1.1. Natural Polymers 

Natural polymers are biodegradable, abundant, and generally non-toxic; they are plant extracts, as well as n-OSA starch and barley protein (Table 1). Natural materials: this category is made up of polysaccharides, proteins, and lipids. Among the most widely used polysaccharides are carrageenan and gum Arabic; examples of proteins are collagen and gelatin; examples of lipids are lecithin and heparin. The most widely used natural materials are polysaccharides, including chitosan and alginate. Chitosan is a polysaccharide with β (1–4) bonds, which can be obtained by alkalinization, and the deacetylation of chitin, naturally found in the exoskeleton of crustaceous. Some advantages of chitosan are its low cost, biodegradability, non-toxicity, and biocompatibility [107].

Alginate is derived from brown algae and is a polyanionic copolymer of D-mannuronic acid and L-guluronic acid linked by b (1–4) bonds. Alginate is widely used as an encapsulating agent for the food industry and the pharmaceutical industry due to its low immunogenicity and biocompatibility [82]. 

#### 4.1.2. Synthetic Polymers 

Synthetic polymers are available in various compositions and various molecules. One of the main advantages of these polymers is that their properties, optimization, and modulation are easy, since they are available in a wide range of compositions and molecular configurations. A disadvantage of these materials is that synthetic polymers show a lack of biocompatibility [82] and are generally biodegradability. Some examples of synthetic polymers are aliphatic polyesters, such as poly (lactic acid) and copolymers of lactic and glycolic acids (e.g., PLGA) [55].

As of 2015, two trends are observed regarding encapsulating materials: the first trend focuses on improving the encapsulation efficiency of conventional materials [33,108,109], and the second trend focuses on the use of novel encapsulating materials [42,59,110]. Both trends can be observed in Table 2, Table 3 and Table 4.

### 4.2. Active Ingredients

#### 4.2.1. Liposoluble Vitamins

Fat-soluble vitamins (A, D, E, and K) have many health benefits. These vitamins are found mainly in food. In pharmacology, these vitamins have been used to treat various diseases, among which are mainly skin diseases, various types of cancer, and the reduction of oxidative stress. These molecules are sensitive to oxidation, so encapsulation could constitute an appropriate means to preserve their properties during storage and improve their physiological potencies. It is interesting that, even when the encapsulation of this kind of vitamin is an important topic, there in not enough. Considering the importance of these types of vitamins and their sensitivity to various factors, there is not abundant information regarding the use of the encapsulation technique by dry spray for the protection of fat-soluble proteins [110]. 

In 2010, the paper “New trends in encapsulation of liposoluble vitamins” mentioned the spray drying technique as a new trend, at that time. This study also highlighted the need for future research and optimization in the field of encapsulation. Although this article covers all fat-soluble vitamins, including carotenoids, only two articles are cited evidencing the use of spray drying for the protection of fat-soluble vitamins: the first article “Modified polyvinyl alcohol for encapsulation of all-trans-retinoic acid in polymeric micelles” in 2005; the second article, “Stability of spray-dried encapsulated carrot” in 1995 [139]. 

##### Vitamin A

Vitamin A is a hydrophobic compound that can aid in the formation and maintenance of healthy teeth, soft and bone tissues, mucous membranes, and skin [139]. Additionally, it is well known that vitamin A is poorly soluble in water due to its low polarity [140]. Vitamin A is also highly sensitive to environmental factors, such as humidity, temperature, light, traces of metals, and heat, among others [141]. It is possible to increase the stability of vitamin A as well as its dispersibility encapsulation using agents with specific physical and chemical properties. Additionally, it is necessary to use the appropriate encapsulation method. This can also benefit vitamin A in determining controlled release kinetics, allowing this vitamin to reach certain sites where absorption is optimized. In this way, both bioavailability and bioaccessibility in the human body are improved [25]. 

Microencapsulation of carotenoids (vitamin A) has been carried out by various techniques such as coacervation [25,142], and emulsion systems [143]; however, spray drying is the most widely used technique [112,144]. Additionally, biopolymers have been reported as encapsulating agents [97]. One of the carotenoid encapsulating agents that has shown advantages in terms of emulsion stability, low viscosity, and good solubility, is Arabic gum [145]. 

Table 2 summarizes the main advances in the use of encapsulation by spray drying for the protection of vitamin A. During the review of encapsulating agents, no new materials have been found, however, investigations are evidenced that seek to improve the efficiency of encapsulation [111,114], as well as mixtures of various encapsulating agents [111].

##### Vitamin D

Vitamin D participates in the calcium-phosphorus homeostasis of the body. Its sustained deficiency causes rickets in children and osteomalacia in adults. However, in addition to this, in recent years it has been observed that vitamin D influences a significant number of physiological processes, especially in relation to the immune system. Thus, various diseases such as cancer, multiple sclerosis, inflammatory bowel disease, arterial hypertension, and cardiovascular disease, have been associated with low levels of vitamin D [146]. Two relevant forms of vitamin D are D2 (ergochole-calciferol) and D3 [147]. 

The sensitivity of vitamin D towards heat, pH, light, oxidation, and hydrolysis, is a big concern during any formulation development [121]. Additionally, the photosensitivity of vitamin D3 in fortified milk during storage resulting in off flavor development. The enrichment of foods with vitamin D has the main role to produce healthy foods in terms of public health [50].

Microencapsulation by dry spray generates particles that exceed five microns, however, there is a trend marked by nanotechnology leading to the modification of the spray drying system towards the generation of nanoparticles obtained by this technique [148].

The article “Vitamin D microencapsulation and fortification: Trends and technologies” written in 2019 shows the little progress in the encapsulation of vitamin D using the spray drying technique. Even though spray drying uses high temperatures, something undesirable for heat-sensitive compounds, this technique offers great flexibility both in wall materials as well as in the proper handling of equipment variables that would allow the preservation of bioactive compounds, sensitive to heat, and that you want to protect. It is evident that the full potential of spray drying is still fully unexplored for vitamin D encapsulation (Table 3).

Recently, articles on this topic have attracted attention, showing advances in the encapsulation of vitamin D by spray drying. Among the most recent news are: saccharomyces cerevisiae yeast used as a vitamin D capsulant [125]; Casein micelles loaded with vitamin D2 [123]; functional yogurt powder fortified with nanoliposomal vitamin D through spray drying [50]; and the co-encapsulation of vitamins B12 and D3 using spray drying [121].

##### Vitamin E

The term, vitamin E, describes a family of eight related fat-soluble molecules. Of these, alpha-tocopherol has the highest biological activity and is the most abundant in the human body. The name tocopherol derives from the Greek word tocos, which means birth, and pherein, which means to carry [149]. A sufficient intake of vitamin E (alpha-tocopherol) is important since it works as an antioxidant, protecting cells, tissues, and organs against the harmful effects of free radicals, which are responsible for the aging process. These can cause a series of conditions, such as: coronary heart disease; cancer or inflammations; inhibits harmful blood clotting, which can block blood flow; regulates the opening of blood vessels, important for blood to flow smoothly [150].

Due to its antioxidant action, vitamin E could protect against clouding of the lenses of the eyes (cataracts) and a progressive deterioration of the retina at the back of the eye (age-related macular degeneration). These two conditions tend to occur with aging, causing a loss of vision [151]. Clinical studies have shown that people with cancer often have low levels of vitamin E in their blood. In addition, population trials suggest that diets rich in antioxidants, including vitamin E, may be associated with a reduced risk of certain types of cancer [150].

Since researchers believe that oxidative stress contributes to the development of Alzheimer’s disease, antioxidants such as vitamin E may help prevent this condition. The fat-soluble vitamin can easily reach the brain and apply its antioxidant properties [152].

Many bioactive substances are lipophilic, such as vitamin E, showing a poor dispersibility in water; high susceptibility to light, heat, and oxygen; and relatively low bioavailability after ingestion. Encapsulating these substances in hydrophilic matrix can convert them to a water-dispersible form, protect them from external factors, mask their flavors, and enhance their bioavailability as well as applicability in food products (Fangmeier et al. 2019) (Table 4).

For vitamin E, it is possible to observe a high numbers or papers in which the spray drying technique is applied not only for general vitamin E, but for specific vitamin E compounds, too. It is interesting that the production of articles related to the use of dry spray to encapsulate vitamin E coincides with the general statistics (Figure 5) where, as of 2015, a high number of publications is evidenced.

There is a trend towards applied research, especially in pharmacology and nutraceutical. Among the most recent applications are: preparation and evaluation of orally disintegrating tablets [138]; the enhancement of oral bioavailability of vitamin E [136]; microcapsules packed in amber colored glass bottle with stability in vitamin E content [135]; the co-encapsulation of coenzyme Q10 and vitamin E [129,132]; and the controlled release of vitamin E from chitosan/sodium lauryl ether sulfate microcapsules [128,131].

#### 4.2.2. Essential Oils

Essential oils (EO) can be extracted from different types and parts of plants (as medicinal, edible, and aromatic), including nutmeg, cloves, oregano, basil, celery seed, coriander, cinnamon, lemongrass, tarragon, thyme, marjoram, garlic, sage, and rosemary. Essential oils are generally recognized as safe (GRAS), thus allowing their use in food [153,154]. Essential oils are characterized by their pleasant and strong aromas, with antioxidant, antibacterial, and antifungal properties, which are conferred by their content of phenolic and aliphatic compounds [155]. Essential oils contain main components which are found in relatively high concentrations (20–70%), some of these main compounds are carvacrol and thymol (oregano), linalol (coriander), α- and β-thujone and camphor (Artemisia herba-alba), α-phelandrene (dill), menthol, and mentone (mint). Low molecular weight essential oils have become a very important group that includes aromatic components, terpenoids and terpenes, as well as aliphatic components [156].

Recently, the article “Green biopolymers from by-products as wall materials for spray drying microencapsulation of phytochemicals” summarize the growing trend towards the use of new encapsulating materials as well as orange waste fiber, carrot fiber, inulin, and pectin, among others. This article is focused in the encapsulation of phytochemicals by spray drying, however, these green biopolymers have been used for the encapsulation of essential oil, lipids or compound lipids, with growing acceptance [45,110].

On the other hand, several review articles were analyzed, with particular attention to two recent review articles focused on the encapsulation of essential oils by spray drying: “Essential oils microencapsulated obtained by spray drying: a review” [155], and “Microencapsulation of Essential Oils by Spray-Drying and Influencing Factors” [28]. Both documents show the use and importance of spray drying, agreeing that the main fields of application of spray drying microencapsulation are food and pharmacology with growth in some fields such as textiles, cosmetics, and packaging. However, these review articles do not include compound lipids and fat-soluble vitamins that are of great interest today.

In food applications, they are used due to their antimicrobial, insecticidal, and antifungal characteristics. On the other hand, in the pharmaceutical industry, health benefits are sought in various diseases such as cancer [157,158,159], stress, antisicosis [160], obesity [161,162], diabetes, antidiabetic, hypolipidemic, anti-inflammatory [163], tuberculosis [164], the polyneuropathy of hereditary transthyretin [165], and acne [166].

Consumers in some industrial sectors have become increasingly demanding regarding the type and quality of the products offered. Among the industry sectors that have implemented innovative technologies are the personal care, perfumery, and cosmetic industries. In order to meet the demands of consumers, effective, natural, and safe products have been created, focused on improving health, contributing to well-being, and highlighting beauty [167,168]. It is possible to find a large number of products that combine new delivery or delivery systems as well as natural and environmentally friendly ingredients [169]. Essential oils have gained great strength in the cosmetic industry and in perfumery because they are natural ingredients proven health benefits and therefore can be used in creams, soaps, skin creams, and shampoos [170]. Since the impact of microencapsulation in the late 1980s, and thanks to the creation of fabrics with aromas developed by Kanebo Ltd., various microencapsulation techniques have been proposed that allow controlling the release of the active compounds of various formulations, whether they are food, cosmetic, or perfumery [171,172].

Some microencapsulation techniques are more used for certain bioactive compounds, for example, the microencapsulation of flavors is mainly carried out using spray drying or extrusion; however, microencapsulation techniques such as coacervation or lyophilization are also used. An essential aspect of the spray drying technique for industry is its easy scalability. The success of spray drying lies in the knowledge of the different variables of the process and the final properties of the microparticles. Some advantages of spray drying are the relative ease of obtaining microparticles in a relatively simple, continuous, and rapid manner. Additionally, it has low production and processing costs and the equipment is available. This technique has been especially used to microencapsulate volatile materials as essential oils, since they require a short contact time in the dryer [173,174].

#### 4.2.3. Polyunsaturated Fatty Acids (PUFA)

The benefits of polyunsaturated fatty acids have been widely reported in relation to health benefits. PUFAs must be supplied through dietary supplements because the human body does not synthesize them; these are also called essential fatty acids (EFAs). Among the most recognized PUFAs are omega-3 (ω-3) and omega-6 (ω-6), reporting great benefits for health and their contribution in terms of energy and well-being. Among the benefits of PUFAs are their ability to minimize the risk of cardiovascular and neurodegenerative diseases as well as help with diabetes, arthritis, and even some types of cancer [175]. The main sources of PUFA are found on the seabed in algae and fish, while on land they can be found in some plant seeds. PUFAs can be short chain like α-linolenic acid and linoleic acid, or they can be long chain such as eicosapentaenoic acid, docosapentaenoic acid, docosahexaenoic acid, and arachidonic acid [176,177].

The spray drying of emulsions is one of the most used microencapsulation and drying technologies in the food and pharmaceutical industries because the process is flexible, economical, efficient, easy to scale-up, uses easily available equipment, and produces good quality powder. It has been extensively used in the encapsulation of fats, oils, flavors, and oil soluble ingredients (Table 5). The general process of spray drying involves the dispersion of a core material into a polymer solution, forming an emulsion or dispersion, the pumping of the feed solution/emulsion, and the atomization of the mixture and the dehydration of the atomized droplets to produce microcapsules [178].

#### 4.2.4. Structured Lipids

Structured lipids are technological innovations based on the chemical composition of fats, with the aim of generating a great nutritional and technological impact in the future. Most lipids of natural origin have limited applications in their original state due to specific fatty acid and triacylglycerol composition [198]. Although human milk fat and cocoa butter are among the few exceptions with inherently excellent functionality, neither of them can meet the substantial demand in the food industry for infant nutrition or chocolate products because of certain social and/or environmental factors [198]. There are increasing health concerns and/or regulatory restrictions on the dietary ingestion of saturated fatty acids (SFAs) and trans fatty acids (TFAs). Trans fatty acids (TFAs) are unsaturated fatty acids that contain at least one nonconjugated double bond in the trans configuration, resulting in a straighter shape. Most commercially prepared foods contain TFAs. Recent data regarding TFA has implicated this lipid as being particularly deleterious to human health, contributing to problems such as cardiovascular diseases, systemic inflammation, dyslipidemia, endothelial dysfunction, and more recently hepatic and neurodegenerative diseases [199].

Lipid modification has been proved to be a powerful tool for addressing the above challenges, where structured lipids with improved function and/or nutrition can be obtained by changing the fatty acid profiles (for example, chain length, unsaturation level, and positional distribution) of natural fats and oils (Kadhum & Shamma, 2017; Osborn and Akoh, 2002).

Structured lipids (SLs) can be produced enzymatically or chemically via interesterification, acidolysis, and/or esterification processes from the conventional fats/oils in order to improve their nutritional and functional properties (Lu, Jin, Wang, and Wang, 2017; Zhao et al., 2014). Recently, the synthesis of SLs via enzymatic reactions is becoming a popular topic of lipid modification, which uses lipases as reaction catalysts. Lipid modifications enable to combine advantages of both medium chain FAs at sn-1,3 positions and long chain FAs at sn-2 position of one triacyglycerol molecule (MLM-type SLs) that makes lipids functional as they can be designed for the special health requirements of patients and meet the demands of consumers who desire functional food products with health-promoting and disease preventing properties (Akoh, Sellappan, Fomuso, and Yankah, 2002).

The chemical structures and molecular architectures of SLs define mainly their physicochemical properties and nutritional values, which are also affected by the processing conditions. The article “Synthesis, physicochemical properties, and health aspects of structured lipids: A review” explains in detail the obtaining, properties, uses and future trends of structured lipids [198]. However, even though these compounds are sensitive to various factors, the various existing or employed methods for the protection of this type of lipids are not shown.

Examples of commercially available structured lipids include the low- or zero-calorie food products such as olestra (Procter and Gambler, Cincinnati, OH, USA), salatrim (Nabisco, East Hanover, NJ, USA) and diacylglycerol (Kao Cooperation, Sumida city, Tokio, Japan) [200].

Structured lipids have applications mainly in health, aiming to treat or prevent diseases such as diabetes, cancer, and obesity, among others. In the last 20 years, there are few investigations focused on the protection of structured lipids using the spray drying technique; this may be due to the great difficulty that occurs during the spraying process because, normally, emulsions have high value viscosity causing plugging in the spraying process [201].

Although structured lipids have important value on their own, they have also been used as the transporters and protectors of compounds sensitive to various environmental factors, in combination with other encapsulating agents [201].

Some of the most relevant articles where spray drying is used for the protection of structured lipids are shown in Table 6.

The production of structured lipids is a growing topic of research. Due to the importance of structured lipids in health and the modifications that these can undergo due to external factors, it is necessary to protect them using an encapsulation technique. The table shows the main investigations carried out with the purpose of protecting structured lipids using the spray drying technique. In this sense, only seven papers were found. The encapsulating agent most used as wall material was isolated whey protein followed by maltodextrin.

The commonly used inlet temperature was 180 °C (Korma et al., 2019) or higher [202], however, lower temperatures also showed good results [201].

## 5. Conclusions and Future Perspectives

Microencapsulation is one of the most popular technologies in recent years due to its wide application in multiple industries, such as pharmaceuticals, food, health, nutrition, and agri-food, among others. Microencapsulation consists of the protection and isolation of a material of interest or biocomposite through the use of a coating material that isolates the biocomposite from environmental factors. Depending on various factors, such as the encapsulating agent, the agent to be encapsulated, and the microencapsulation technique, it is possible to obtain varied morphologies as well as to modify the performance of the microparticles. In pharmacology, microencapsulation is widely used in processes such as controlled release and prolonged release. In the case of the food industry, microencapsulation is used for the elaboration of functional foods protecting bioactive compounds, such as enzymes or vitamins. In medicine, it has been used for cell immobilization mainly for cell encapsulation for use in regenerative medicine and tissue engineering.

Each microencapsulation technique has particular characteristics that allow them to be applied more efficiently and effectively in certain areas and certain processes. The foregoing implies that each microencapsulation technique has both advantages and disadvantages since microencapsulation techniques differ in parameters such as particle size, encapsulating material used, particle morphology, release kinetics, stability and useful lifetime of the bioactive component, as well as ease of handling and production costs. Regarding the associated costs, it is of great importance to consider each microencapsulation technique used since the scaling processes or their simulations are vital before deciding to implement a certain technique.

Microencapsulation is widely used for the protection of compound lipids; however, despite the fact that many microencapsulation techniques have been used, spray drying is the most widely used encapsulation technique for the protection of this type of compound. Spray drying has been used to encapsulate essential oils, fat-soluble vitamins, and polyunsaturated fatty acids. Spray drying is a technique that is easy to implement, has good encapsulation performance, is inexpensive, and is easy to scale. Additionally, this technique is compatible with the use of a large number of encapsulating materials, including starches, maltodextrins, gums, and sugars.

In short, microencapsulation is a growing technique. Within encapsulation techniques, spray drying stands out as it offers wide advantages for various applications in different industries. However, there is the challenge of optimization in terms of the percentage of performance of this technique, as well as the great challenge of responding to the demands of consumers regarding the protection of bioactive compounds of lipid origin that respond to nutritional needs, food, and health.

Future trends in the field of microencapsulation will likely focused on: (i) the implications for the food industry, medicine, cosmetic, and textile industries based on different lipidic bioactive compounds aiming to create a synergistic effect; (ii) it is important to deepen the encapsulation of structured lipids due to their contributions to health, and (iii) the stability and bioavailability into the gastrointestinal system.

## Figures and Tables

**Figure 1 pharmaceutics-15-01490-f001:**
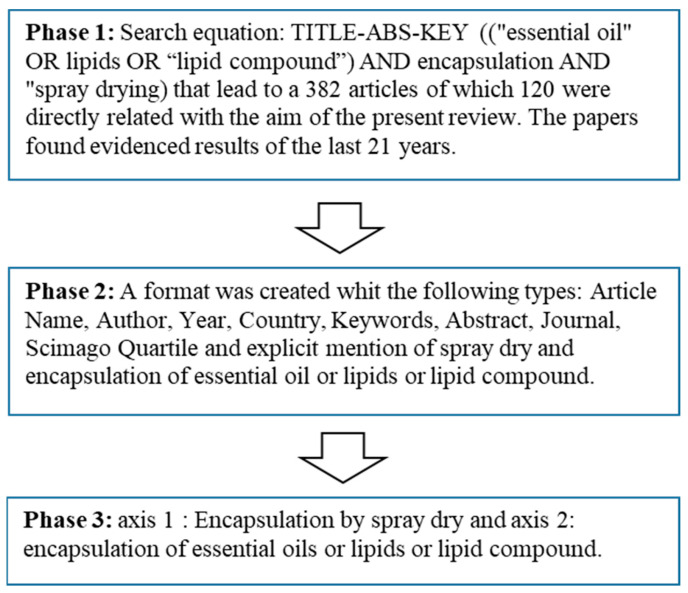
Methodology outline.

**Figure 2 pharmaceutics-15-01490-f002:**
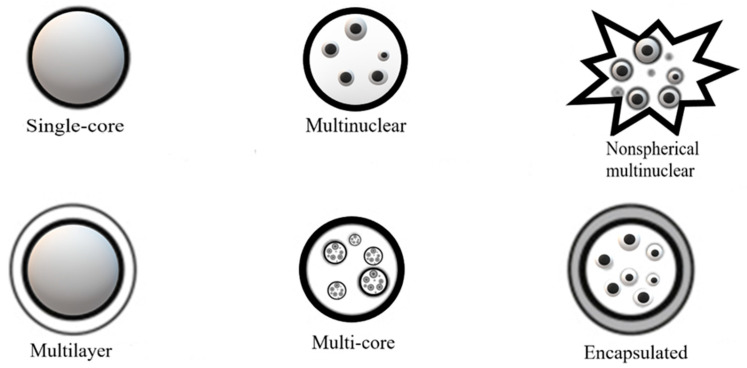
Morphologies of particles obtained using encapsulation processes (adapted from [17]).

**Figure 3 pharmaceutics-15-01490-f003:**
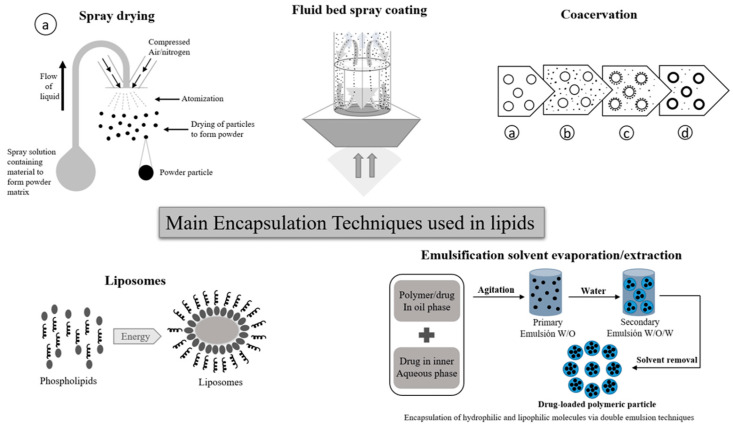
Encapsulation techniques used for lipid protection.

**Figure 4 pharmaceutics-15-01490-f004:**
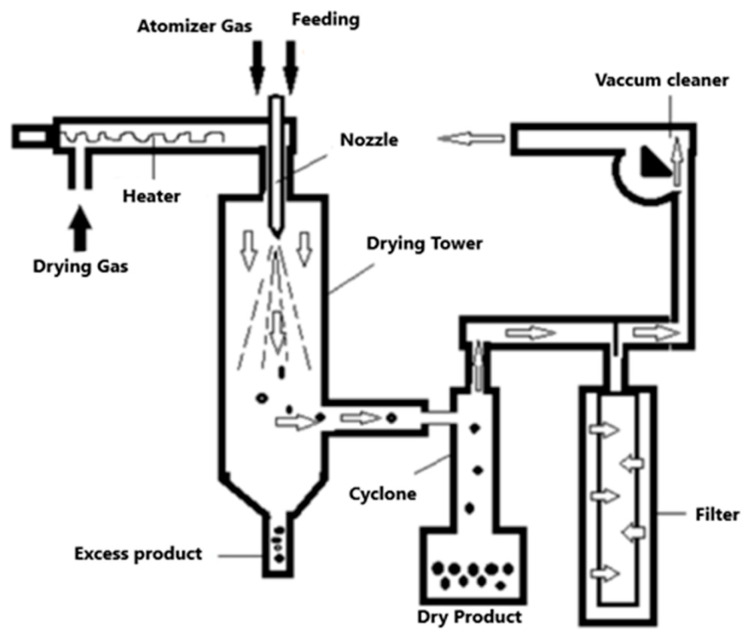
Spray drying encapsulation process.

**Figure 5 pharmaceutics-15-01490-f005:**
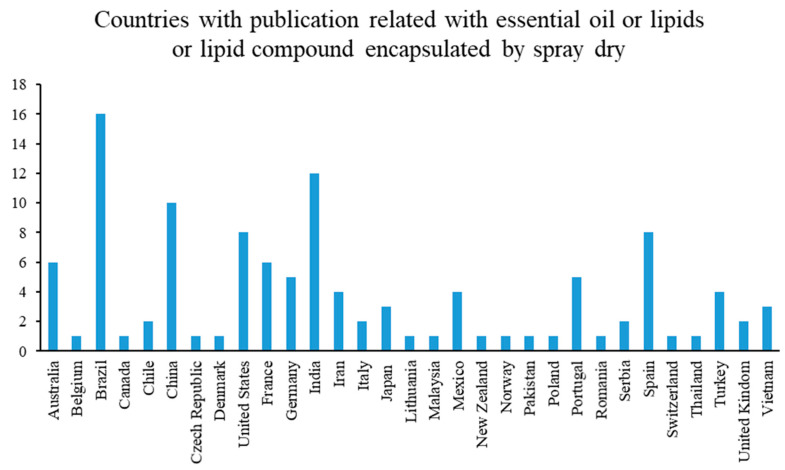
Countries and publications.

**Figure 6 pharmaceutics-15-01490-f006:**
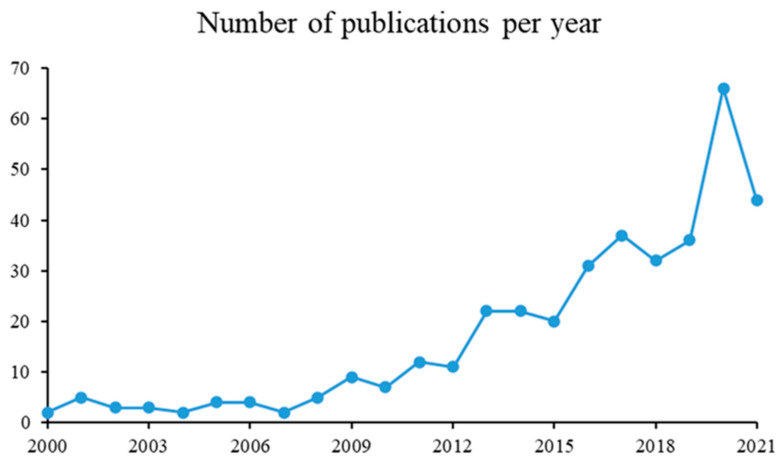
Number of publications per year.

**Figure 7 pharmaceutics-15-01490-f007:**
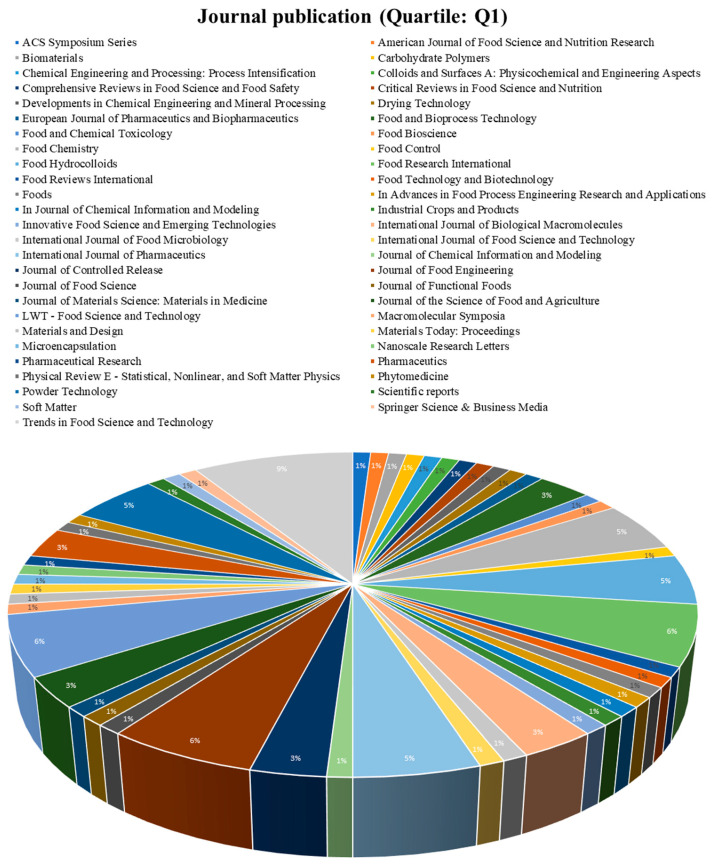
Journal publication (Quartile: Q1).

**Figure 8 pharmaceutics-15-01490-f008:**
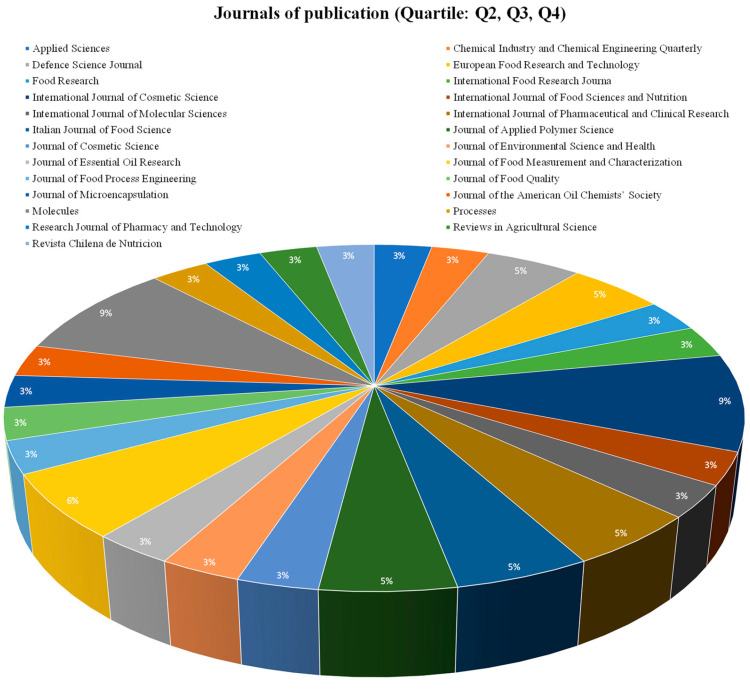
Journal publication (Quartile: Q2, Q3, Q4).

**Table 1 pharmaceutics-15-01490-t001:** Common encapsulating agents used in lipids (years 2000–2015).

Encapsulating Agent	Reference
Barley protein	[83]
Carrageenan, rice bran	[84]
Corn syrup: Sodium caseinate: Lecithin	[85]
Glucose syrup	[86]
Gum arabic and modified starch	[87]
Gum arabic and modified starch (CAPSUL and HI-CAP 100)	[88]
Gum Arabic or maltodextrin 20 dextrose equivalent	[89]
Gum arabic/starch/maltodextrin/inulin	[90]
Hydroxypropyl methyl cellulose, maltodextrin, and, colloidal silicon dioxide	[91]
Maltodextrin	[92]
Maltodextrin: Fish gelatin: k carragenan	[93]
Maltodextrin: n-OSA starch:Whey protein concentrate	[94]
Methylcellulose: Calcium-gelatin casein: Whey protein concentrate Maltodextrin Soy lecithin	[95]
Methylcellulose: Maltodextrin: Lecithin	[96]
Modified starch (Capsul^®^)	[97]
n-OSA starch: Glucose syrup	[98]
Powder milk (SMP) and whey protein concentrate (WPC)	[99]
Skim milk powder: Whey protein concentrate: Whey protein isolate: Milk protein concentrate: Sodium caseinate	[100]
Soy fiber: Maltodextrin: Hydroxypropyl bcyclodextrin: n-OSA starch	[101]
Sugar beet pectin: Glucose syrup	[102]
Sugar beet pectin: Glucose syrup	[103]
Whey protein and maltodextrin	[104]
Whey protein isolate	[105]
Zein/casein complex	[106]

**Table 2 pharmaceutics-15-01490-t002:** Vitamin A encapsulation by spray drying: recent articles.

Encapsulated Material	Encapsulation Efficiency	Essential Oil, Lipids or Lipidic Compound	Encapsulation Conditions	Principal Results	Reference
Gum arabic, maltodextrin and starch	88–98%	Vitamin A	In the encapsulation process, 150 °C was established as the inlet temperature. Air pressure and aspiration rate were set to 5–6 bar and 100% (36 m^3^/h), respectively.	It is possible to prove the viability and integrity of the vitamin A particles produced in the current work, which revealed great encapsulation efficiency values and, at the same time, a total release of the active compound when placed in the proper medium.	[111]
Capsul-CAP^®^, sodium caseinate-SC in combination with Tween 80 (TW) as an emulsifier and maltodextrin (MD)	Vitamin A: 23–100%; Vitamin E: 29–48%	Vitamin E and Vitamin A	Nozzle air flow-rate of 1.052 m^3^/h and aspiration of 80% (32 m^3^/h). The inlet and outlet temperatures were 120 ± 1 °C and 74 ± 1 °C, respectively.	The proposed encapsulation methodology is, therefore, a feasible alternative for the stabilization of vitamin A and E and protection against oxidation processes in the feed manufacturing industry.	[112]
Gelatin	-	Vitamin A acetate (VA)	Inlet temperature = 120–140 °C, feeding rate = 20 mL/min and solid content = 25–30%.	The optimized emulsion conditions are: mass ratio of gelatin: VA = 4:1; emulsion temperature = 60 °C, emulsion pH = 4.5, emulsion time = 40 min and shear rate = 10,000 r/min.	[113]
Sodium caseinate and pea protein	95–98%	Vitamin A	The inlet air temperature of the dryer was set for five different levels from 187 °C to 127 °C, and the outlet temperature was at 80 °C. The targeted moisture content of final spray-dried powder was set at 2.5%.	The results also indicated a potential inherent correlation between properties of liquid emulsion and powdered microcapsules. On the other hand, a lower spray drying inlet temperature at 127 °C increased the moisture content and water activity and decrease the glass transition temperature of spray-dried powders, which consequently resulted in powder caking and nutrients degradation.	[114]
Arabic gum	5.1 and 33.9%	*β*-carotene (Precursor or vitamin A)	The drying air flow rate was set at 47 m^3^/h. The feed solution was kept under magnetic stirring. The pressure of the compressed air set at 1.7 bar and had a maximum flow rate of 73 m^3^/h. The inlet temperature ranged between 110 and 200 °C.	The drying inlet temperature of 173 °C and the Arabic gum concentration of 11.9% were those that allow obtaining higher *β*-carotene content, higher encapsulation efficiency, and higher drying yield.	[115]
Poly(D,L-Lactide–co-glycolide) (PLGA)	70.5 ± 2.3%	Metabolite of vitamin A	The spray dryer was operated using Nitrogen at 670 L/h (55 mm), an aspirator rate of 100%, an inlet temperature of 50 °C and a solution feed rate of 4 mL/min (15%).	The results also show the benefit of all-trans-retinoic acid (ATRA) as a practical treatment post-infection, and high light the importance of appropriate nutrition in host-protective immune responses to tuberculosis disease.	[116]
Gum arabic and maltodextrin	44.1%	Vitamin A (shark liver oil)	Inlet temperature, 150 °C; outlet temperature, 90 °C; air flow rate, 600 L/h; and drying air flow rate, 60 m^3^/h.	Best encapsulation efficiency and moisture content for its conservation, the combination of gum Arabic and maltodextrin, as encapsulation agents, should be maintained at 47% and 23%, respectively.	[117]
Maltodextrin	-	Vitamin A	Inlet temperature: 90–120 °C	Spray drying, combined with a dehumidifier and a double condenser to test vitamin A concentrations in a mixture of tomato juice and maltodextrin, can be operated up to a temperature of 90 °C.	[118]
Arabic gum	-	Vitamin A (Retinol)	The air and solution flow rates, air pressure, inlet and outlet temperature were set at 35 m^3^/h (90%), 3–6 mL/min (between 10 and 20%), 5–6 bar, around 150 °C and around 88 °C	Vitamin A release assays showed that the usage of 2, 5 and 10% (*w*/*v*) of Arabic gum do not ensure an efficient protection and stabilization of vitamin A. It was necessary to increase the encapsulating agent concentration until 15 and 20% in order to obtain the release of initial amount of vitamin A used in the assays.	[119]
HI-CAP 100 (starch octenylsucciniate, OSA-starch)	96.38%	Vitamin A	The emulsions were spray-dried at a feed rate of 1000 mL/min. The optimum air inlet and outlet temperature were 182 and 82 °C, respectively	Vitamin A microcapsules produced with HI-CAP 100 exhibited spherical shapes with characteristic dents, which was attributed to drying and cooling solidification involved during spray-drying. The vibrating frequency of the centrifugal granulation had effect on the particle size distribution of microcapsules (*p* < 0.05).	[120]

**Table 3 pharmaceutics-15-01490-t003:** Vitamin D encapsulation by spray drying: recent articles.

Encapsulanting Agent	Encapsulation Efficiency	Essential Oil, Lipids or Lipidic Compound	Encapsulation Conditions	Principal Results	Reference
Gum acacia: Hi-Cap^®^ 100: maltodextrin = 38:60:2	77–93%	Vitamin D3	The aspirator and feed rate set at 1400 rpm and 20 rpm, respectively, and compressed air flow pressure was adjusted to 2 bar.	A wall material combination of 38:60:2 ratio of gum acacia, Hi-Cap^®^ 100, and maltodextrin showed best physico functional properties for co-encapsulation of vitamins B12 and D3 as seen from the physico functional parameters such as entrapment efficiency, total encapsulation efficiency, and process efficiency of the microcapsules as well as the storage and thermal stability of the vitamins entrapped therein.	[121]
Chitosan/ethylcellulose	95%	Vitamin D2	Solutions were spray-dried at a feed rate of 5 mL/min. The air inlet temperature was 168 °C and pressure 0.38 MPa.	The drug loading of this system was more than 86%.	[122]
Casein micelles (CM)	88%	Vitamin D2	The inlet air temperature was 180 °C and the outlet air temperature was 80 °C. The suspension was pumped with a gear pump which operated at 285–410 rpm. between 191 and 203 g/min, approximately 6.7 kg per minute.	The recovery rates for Vit. D2 were 76% (spray-dried powders), The Vit. D2 content stayed constant in all powders during four months of storage, 90% of the Vit. D2 added as encapsulated product in dried CM remained active after in vitro proteolysis.	[123]
Maltodextrin (MD), gum Arabic (GA), modified starch (MS), and whey protein concentrate (WPC)	96.4%	Vitamin D3	Air gauge pressure was kept at 0.06 MPa and air flow rate at 73 m^3^/h. The inlet temperature was set at four different levels (160, 170, 180, and 190 °C) and outlet temperature at 80 ± 5 °C.	Vitamin D3 was encapsulated into nanoliposomes and then formulated by biopolymers including MD, GA, MS, and WPC. Finally, the produced feed solutions were turned into powders through spray drying. The results showed that the inlet air temperature and carrier agents had a significant effect on whey powder characteristics loaded with nanoliposomal vitamin D3.	[124]
Milk protein concentrate, modified starch content, gum Arabic and maltodextrin	-	Vitamin D	Feed mixture was atomized from the nozzle into a vertical co-current drying chamber with 2 m height, while the hot air flowrate, atomizing air pressure and outlet air temperature were fixed at 550 L/h, 0.3 bar and 90 °C. For all experiments, the type and concentration of drying air as well as inlet air temperature (160, 170, 180 or 190 °C) were independent variables.	Properties of yogurt powders fortified with encapsulated vitamin D are significantly dependent on drying conditions and feed mix ture, and achieving favorable properties of fortified yogurt powders is made possible through optimization of independent variables.	[50]
*Saccharomyces cerevisiae yeast cells*	-	Vitamin D	Inlet temperature 130 °C, outlet temperature 75–77 °C, feed flow rate 6.08 mL/min, nozzle diameter 0.7 mm, dry air flow rate 568 L/h, aspirator 90% and pump rate 25%.	Yeast based microencapsulation technique was used success-fully for encapsulation of cholecalciferol. The *Saccharomyces cerevisiae* yeast cell microcapsules could serve as a novel carrier for encapsulation of cholecalciferolin order to increase its bioavailability for using in food and pharmaceutical industries.	[125]
Ovalbumin	-	Vitamin D	Inlet temperature 60 °C, outlet temperature 35 °C, aspiration 85%, feeding rate of the suspension 5 mL/min.	The result revealed that VD-loaded nanoemulsions (VDNM) led to an improvement in oral bioavailability (BA) of Vitamin D in amurine ovalbumin-induced asthma model. These data provided an important proof that VDNM might be a new potential therapy for the management of asthma in humans.	[126]
Trehalose–maltodextrin and lactose–maltodextrin	-	Vitamin D3	The operational conditions of the spray drying were air inlet temperature: 120 °C and flowrate: 51.4 mL/min.	β-lactoglobulin (β-LG) has been reported to be capable of binding a variety of fat-soluble ligands, including vitamin D3. The importance of the binding property is that it is possible to deliver vitamin D3 using β-LG as a carrier without the presence of the fat in which it normally associates.	[127]

**Table 4 pharmaceutics-15-01490-t004:** Vitamin E Encapsulation by spray drying: recent articles.

Encapsulating Agent	Encapsulation Efficiency	Essential Oil, Lipids or Lipidic Compound	Encapsulation Conditions	Principal Results	Reference
Chitosan and sodium lauryl ether sulfate (SLES)	73%	Vitamin E	Aspiration (0.6 m^3^/min) and feeding (2.2 mL/min). Inlet temperature 160 °C and the outlet temperature 100 °C.	The use of aldehydes as cross-linking agents and found that chitosan/SLES complex can be used as wall material for the microencapsulation of hydrophobic active molecules in cosmetic industry.	[128]
Whey protein isolate (WPI), WPI/soluble corn fiber (SCF), and WPI/maltodextrin	87.4 and 91.0%	Vitamin E with coenzyme Q10	Nozzle 100-μm and spray drying temperature was 190 and 90 °C for the inlet and outlet, respectively.	The composition and property of wall material governed most powder properties and influenced some important functionalities such as proneness to digestion-induced disintegration. Core material impacted on particle morphology and color and played a key role on stabilizing powder functionalities during storage.	[129]
Ethylcellulose (EC)	21.60 and 99.75%.	Tocotrienol (vitamin E compound)	Inlet temperature 80–90 °C, outlet temperature 70–80 °C; feed flow 5 mL/min; pressure 3 bar.	The microencapsulation of tocotrienol with EC using SE (Solvent Evaporation) and spray drying techniques produced a solid form of tocotrienol that was considerably more stable than the natural form of tocotrienol.	[130]
Capsul-CAP^®^, sodium caseinate-(SC) in combination with tween 80 (TW) as an emulsifier and maltodextrin (MD)	Vitamin A: 23–100%; Vitamin E: 29–48%	Vitamin E and vitamin A	Nozzle air flow-rate of 1.052 m^3^/h and aspiration of 80% (32 m^3^/h). The inlet and outlet temperatures were 120 ± 1 °C and 74 ± 1 °C, respectively.	The proposed encapsulation methodology is therefore, a feasible alternative for the stabilization of vitamin A and E and protection against oxidation processes in the feed manufacturing industry.	[112]
Carboxymethyl starch (H-CMS) and xanthan gum (XG)	57–67%	Vitamin E	The inlet and outlet temperatures of spray-drying were 190 ± 5 °C and 80 ± 5 °C, respectively.	H-CMS may be used to construct a pH-sensitive functional 399 factor delivery system, which further expands its practical application and has a certain guiding 400 significance for the use of starch in the production of value-added products.	[131]
OSA (octenyl succinic anhydride) modified starch (HICAP100)	98–99%	Coenzyme Q10 (CoQ10) and vitamin E (VE)	Inlet/outlet temperatures of 160/70 and 190/90 °C, respectively. Airflow rate was set at 250 L/min, with a feed rate of 1.5 mL/min.	The CoQ10 and VE retention, antioxidant capacities and color of the microcapsules were relatively stable when spray-dried at 190 °C than at 160 °C.	[132]
Maltodextrin and sodium caseinate	60–71%	Vitamin E	Inlet temperature 110 °C, air pressure 55 kgf/cm^2^, and atomizer speed 20,000–25,000 rpm, nozzle 1.5 mm.	The best core/wall ratio obtained in this experiment is 1.0 for its efficiency and physical characteristic although it showed the tendency of agglomeration.	[133]
Gum acacia (GA) and mixed of galactomannan from Arenga pinnata (GAP) with GA	60–70%	Vitamin E	Initial temperature 70 °C for 15 min. Inlet temperature (180–200 °C).	The increment of GAP decreasing moisture content and the particle size from 16 μm to 11 μm, the yield of microcapsule, encapsulation efficiency, the amount of vitamin E absorbed and oxidation stability of vitamin E were increased.	[134]
Cremophore RH 40, tween 80, maltodextrin, OSA-modified starches (Capsul and Hicap100)	53–63%	Vitamin E acetate	Inlet and outlet temperatures were 110–130 °C and 55–60 °C, feed rate 1–5 mL min^−1^, atomization air pressure 2–3 kg cm^−2^ and aspiration rate 40–45%.	The microcapsules packed in amber colored glass bottles exhibited no significant change in moisture content and drug content indicated microcapsules were stable for 3 months at accelerated conditions.	[135]
Whey protein	89%	Vitamin E	Inlet and outlet temperatures 100 °C and 80 °C, respectively. The feed liquid flow rate 4 mL/min.	It was demonstrated that pharmacokinetic parameters were improved using the spray freeze drying technique over that of spray drying and freeze-drying techniques. The spray freeze-dried vitamin E microcapsules were able to increase the oral bioavailability by 1.13 and 1.19-fold compared to spray dried, and freeze-dried microcapsules respectively. Thus, this study indicated that spray freeze drying technique could be potentially employed for encapsulating poorly water-soluble bioactive compounds.	[136]
Octenyl succinic anhydride (OSA) modified starches	-	Vitamin E	Inlet and outlet temperatures were 150 °C and 85 °C respectively, feed rate 10 mL/min.	This study might be useful to service providers interested in delivering Vitamin E in form of nanocapsules in identifying appropriate modified starch to act as emulsifier and wall material.	[137]
Hydrolyzed gelatin	-	Vitamin E (VE) (d-α-tocopheryl acetate and d-α-tocopheryl acid succinate)	Inlet temperature 200 °C, outlet temperature 100 °C, and the feeding rate 1.5 mL/min.	Tablet porosity of 30 to 35% and tensile strength of 7 kg/cm^2^ or greater are required for VE orally desintegrating tablets (ODTs) to rapidly disintegrate and have sufficient strength. It has also been demonstrated that, for the addition of VE, VE spray drying granules of small particle size and powder VE are the most suitable.	[138]

**Table 5 pharmaceutics-15-01490-t005:** Essential oils and PUFA Encapsulation by spray drying: recent articles.

Encapsulated Material	Encapsulation Efficiency	Essential Oil, Lipids or Lipidic Compound	Encapsulation Conditions	Principal Results	Reference
Rice and whey protein	40–50%	Baltic herring (BH) oil	Inlet air temperature in the range of 123–129 °C, and outlet temperature in the range of 72–78 °C.	Production of emulsions with BH oil and whey protein concentrate and rice protein concentrate (RPC) mixture as wall material components, resulted in stable emulsions with relatively small droplet size and large dispersion. However, while RPC was shown to either agglomerate or stay non-dissolved at pH 3, but surprisingly, at pH 3, the most stable emulsion was obtained.	[179]
Hydrolyzed sunflower lecithins, chitosan and chia mucilage	84.11–99.37%	Chia seed oil	Feed rate of 0.6 L/h, and air inlet/outlet temperatures of 170 and 75 °C, respectively	Chia oil microcapsules with appropriate physicochemical stability were obtained through the spray drying of multilayer emulsions pre-pared using adequate electrostatic deposition by the layer by layer technique. A high microencapsulation efficiency was obtained, suggesting that the type and concentration of wall materials were suitable in trapping and containing the lipid nucleus.	[180]
Octenyl succinic anhydride–linked starch (OSA-S) and maltodextrin (MD)	Not reported	Fish oil	Inlet air temperature 180–190 °C, outlet air temperature 80–90 °C, feeding speed 20 mL/min, and atomizer speed 200–300 r/min	The microcapsules were not resistant to acid treatment and had a lower oxidation rate in neutral condition. Moreover, the results of in vitro digestion investigations showed that the fish oil microcapsules were easily dissolved and released in simulated gastric fluid, which was also confirmed with confocal laser scanning microscopy (CLSM).	[181]
Low-molecular-weight keratin (LMWK)	Not reported	Fish oil	The diameter of the feed nozzle was 0.75 mm and the air pressure was 0.6 bar. The inlet and outlet temperatures were 175 and 80 °C, respectively.	In the present work, LMWK was successfully applied as part of the wall material during the spray drying process for fish oil encapsulation. Under the same drying conditions microcapsules containing LWMK illustrated lower moisture content and higher encapsulation efficiency and anti-ultraviolet capability. The beneficial effects of LMWK were enhanced with increasing proportionality.	[182]
Sodium caseinate and lactose	58.8–76.9%	Omega-3 (lipids from oil seeds and microalgae)	Air inlet temperature 170 °C, compressed air pressure 5 bar, air flow 700 L/min and aspiration 70%	Microencapsulation efficiency depended on the type of lipid extract to encapsulate and varied from 57.0 to 76.9%. The highest microencapsulation efficiency was found for chia fatty acid ethyl esters microcapsules (76.9%), while echium microcapsules showed the highest payload (142 mg/g).	[183]
Konjac glucomannan (KGM) and soybean protein isolate (SPI)	90.10%	Fish oil	The pump rotation speed at 20 mL/min, the temperature of the air at the inlet and outlet of the dryer were 200 and 80 °C, respectively	Release kinetics test further indicated retention rate of core materials for microcapsules prepared with spray drying were better than with freeze-drying. In addition, a human epithelial microfold cell (M-cell) transcytotic assay demonstrated that the M-cells had greater transport activity for the exogenous microcapsules.	[184]
Whey protein isolate (WPI) and octenylsuccinic anhydride (OSA) modified starch	94.0–95.1%	β-carotene, lutein, zeaxanthin, and fish oil	The flow pressure was 0.4 psi, inlet temperature was 180 °C, and outlet temperature was controlled in a range of 85–90 °C	This study has provided an alternative way of delivering visual-beneficial compounds via a novel drying method, which is fundamentally essential in both areas of microencapsulation application and functional food development.	[185]
Gelatin, gum Arabic and maltodextrin	83–95%	Fish oil	The inlet air temperatures were 190 ± 2 °C, and the feed flow rate of the emulsion was 3 mL/min, leading to the recorded outlet air temperature of 60 ± 2 °C	The microcapsules prepared by coacervation of gelatin and gum Arabic followed by spray coating with a mixture of gelation and maltodextrin were the strongest and stiffest based on the calculated nominal rupture stress and Young’s modulus, respectively.	[186]
Maltodextrin and modified starch	69–87%	Fish oil	VSD process is carried out under low evaporation temperature (around 30 °C) and airflow (only atomization air of 20 L/min	The oxidative stability of the oil was greater in the vacuum spray drying (VSD) particles confirmed by Rancimat and Oxipres methodologies. Regarding the consolidation of VSD as a commercially competitive dryer, modifications must be made to your project with the aim of improving the transfer of heat and mass and achieving at least feed rate ranges similar to those employed in the spray drying.	[108]
Whey protein isolate, gelatin and Capsul^®^	42.5–94.6%	Unsaturated triglyceride (fish oil) and (orange essential oil)	Iinlet and outlet air drying temperature of 180 °C and 90 ± 3 °C, respectively	The interfacial membrane surrounding the oil droplets is suggested to be determinant in the oxidative stability. The protein matrices showed antioxidant capacity that also can contribute to high protection.	[187]
Hydroxypropyl-inulin (HPI)		Fish oil (FO)	The inlet gas temperature was from 150 to 200 °C (conventional spray drying) and from 75 to 135 °C (water- free spray drying	FO-conventional spray drying and FO-water-free spray drying microparticle systems showed encapsulation efficiency values of FO above 80%, in spite of the different FO encapsulation mechanism (emulsion retention and triglyceride-HPI interactions, respectively). However, the type of solvent slightly affected the microparticle properties (Tg, moisture, hygroscopicity, FO release and FO oxidative stability)	[188]
Maltodextrin and soy protein isolate	90–94%	Fish oil	The airflow rate was set to 250 kg/h and emulsions dried at an inlet and outlet temperature of 180 and 87 °C, respectively	Due to the standardization of the particle size and the determination of oxidation products in the total- and encapsulated oil, the influence of size and non-encapsulated oil could be eliminated. The oxidation of encapsulated lipids is limited by the oxygen availability and supply rather than by the oil load. This is explained by two effects, the oxygen diffusion and a scavenging activity of the oil located in the outer particle region consuming the penetrating oxygen and thereby protecting oil droplets in the particle center.	[189]
Gum Arabic and maltodextrin	Not reported	Carotenoids	6 bars air pressure and 740 L/h pressured gas flow feed. In respect to the airflow and the inlet and outlet temperatures of drying air were at a first trial 160 ± 2 °C and 70 ± 2 °C, respectively	The results related to Individual carotenoids content of the microcapsules, however, presented a considerably diminished lycopene content after atomization. Furthermore, undetectable quantities of β-carotene were observed in the gastric phase of the simulated digestion of the microcapsules indicating a strong degradation process in the acidic environment.	[109]
Deoiled or hydrolyzed sunflower lecithins, chitosan and chia mucilage	84–99%	Chia oil	Feed rate of 0.6 L/h, and air inlet/outlet temperatures of 170 and 75 °C	All the microcapsules studied were efficient to protect chia oil against lipid oxidation (<10 meq hydroperoxides/kg oil), mainly the three-layer ones. The omega-3 PUFAs content after storage presented the highest levels in the three-layer microcapsules and decreased only in the monolayer system.	[180]
Maltodextrin and modified starch	69–87%	Fish oil	The fresh emulsion was fed into the drying chamber at a 0.012 L/min	Particles had a lower mean diameter (6.9 μm) when compared to spray drying particles (14.6 μm), which favors the reduction of occluded oxygen. Both samples showed a continuous wall with no apparent cracks, which is an important factor to provide better protection of active. The oxidative stability of the oil was greater in the vacuum spray drying particles confirmed by Rancimat and Oxipres methodologies.	[108]
Macadamia protein isolate (MPI) and chitosan hydrochloride (CHC)	94.2%	Macadamia oil	Flow rate of 5 mL/min. The aspirator was set at 100%, the actual air flow rate was 538 L/h, the inlet air temperature was set at 160 °C and the outlet air temperature at 85 ± 2 °C. Macadamia oil powders were collected and stored at 4 °C before being analyzed.	Optimum MPI/CHC level of 5:1 for producing the macadamia oil microcapsules because this gave a high encapsulation efficiency, strong protection against lipid oxidation, and good storage stability after rehydration.	[190]
Gum Arabic (GA), whey protein isolate (WPI), maltodextrin (MD)	82.34–87.19%	Basil essential oil (BEO)		Finally, the GA:WPI:MD formulation demonstrated a high product yield and encapsulation efficiency with better physicochemical properties for encapsulation of BEO.	[61]
Maltodextrin, gum Arabic and whey protein	92.80–97.38%	Seed oil	The temperature of inlet air was maintained at 180 ± 1 °C, the outlet temperature was 80 ± 1 °C, and the direction of hot air was co-current. The atomizing air inlet speed was 3 m^3^/h, while the feeding speed was 20 mL/min	Carbohydrate-based microencapsulation showed the highest relative crystallinity, the temperature of the glass transition (Tg), which indicated good stability. Carbohydrate-based microencapsulation greatly improved the oxidative stability of gurum seeds oil suggesting better safeguarding of this sensitive oil.	[191]
Hydroxypropyl (HP) α-, HP β- and HP-γ-CD, cyclodextrins (CDs)	Carvacrol (95.7–98.1%), Thymol (70.2–79.8%)	Carvacrol and thymol	Inlet air temperature, 170 °C; outlet air temperature, 68 °C; nlet air flow 35 m^3^/h, pump flow 5 mL/min	Spray-drying method, mainly combined with HP-γ-CD, allows for obtaining solid complexes that maintain an antimicrobial activity of a level comparable to that displayed by compounds in a free state.	[192]
Maltodextrin, SAC (whey protein isolate, gelatin or Capsul^®^)	48–95%	Fish oil, orange essential oil (OEO)	Inlet air temperature, 180 °C; outlet air temperature, 90 °C.	The protein matrices showed antioxidant capacity that also can contribute to high protection. This work provided insights about the understanding of how barrier properties of powders affect oxidation.	[187]
Gum Arabic (GA), maltodextrin (MD), and whey protein isolate (WPI)	82.34 and 87.19%	Basil (*ocimum* *basilicum* L.) essential oil (BEO)	Inlet air temperature, 150 °C; feed rate of 3 mL/min, drying air flow rate of 40 Kg/h	The GA:WPI:MD formulation demonstrated a high product yield and encapsulation efficiency with better physicochemical properties for encapsulation of BEO.	[61]
Maltodextrin	92%	Lemongrass (cymbopogon citratus) essential oi	Inlet temperatures: 60, 100, 140, and 180 °C	When the temperature and the time increased, the color of powder became dark and OR values were rapidly reduced. The selected optimal temperature and time was 100 °C and 20 min.	[33]
Bovine serum albumin, gum acacia and oxidized starch crosslinker	Not reported	Peppermint Oil	The inlet air temperature was varied from 135 to 145 °C and the aspirator rate was maintained at 100%	After complex coacervation, all the reactant ratios used here resulted in stable spherical mononuclear core-shell capsules, with no measurable loss of peppermint oil compared to the parent emulsion. Samples without crosslinker did not withstand spray drying, thus demonstrating the need for reinforcing the complex coacervate walls with a cross linker or with a common additive such as a sugar.	[48]
Chitosan tween-80	39%	Lemongrass essential oil	Temperature input 160–165 °C	Chitosan microparticles loaded with essential oil (CMEOs) had higher thermal stability and presented better colloidal stability than chitosan microparticles and pure oils. The results also demonstrated that the proposed system allows controlled release of the bioactive com-pound.	[193]
Octenyl succinic anhydride–linked starch (OSA- S) and maltodextrin (MD)	Not reported	Fish oil	Inlet air temperature 180–190 °C; outlet air temperature 80–90 °C; feeding speed 20 mL/min; and atomized speed 200–300 r/min.	Different temperatures and pH had significant effects on the oxidation stability of fish oil microcapsules as observed an upward trend for peroxide value, acid value, and thiobarbituric acid test during storage. The microcapsules were not resistant to acid treatment and had a lower oxidation rate in neutral condition.	[181]
Artichoke bracts flour, maltodextrin, tween 20	65–79%	Sunflower oil	Inlet air temperature 175 °C; outlet air temperature 100 °C aspirator 80%	Artichoke bracts are a healthy alternative to synthetic emulsifiers and could be successfully combined with common wall materials for lipids microencapsulation.	[194]
Maltodextrin and gum Arabic	87%	Chia oil	Inlet air temperature 100–120 °C; drying airflow of 1.65 m^3^/min; air atomizing pressure of 4 bar	Treatment 4 (120 °C for inlet temperature and 0.1 L/h of feed rate) is the most indicated for application in food products.	[195]
*S. edule* fruit starch (SS) in combination with whey protein (WPC) and gum Arabic (GA)	60–78%	Cinnamon oleoresin	The drying conditions were inlet/outlet air drying temperatures of 150/85 °C, using a parallel arrangement of the nozzle concerning the drying airflow, and a nozzle pressure of 2.5 kPa. Throughout the drying process, the emulsion was continuously agitated	Cinnamon oleoresin microcapsules synthesized with ternary formulations stabilize the phenolic compounds in cinnamon oleoresin and reach higher encapsulation efficiency values. Based on these results, we suggest that *S. edule* fruit can be used as a starch source. When combined with other wall materials.	[196]
Biopolymers, gelatin and lignin	97.0%	Orange essential oil	Inlet air temperature 110–150 °C; Feed flow rate 0.15–0.45 L/h; Drying air flow 301–536 L/h	It was possible to prepare microparticles showing microspherical morphology, with the average particle size of less than 4 μm (gelatin) and 3 μm (lignin), with an oil content of 90% (*w*/*w*) for gelatin and powder recovery of 72% (*w*/*w*) for lignin as wall material.	[110]
Whey protein isolate (WPI) as the primary wall material by prebiotic carbohydrates, such as maltodextrin (MD) and inulin (IN)	89.10%	Structured lipids (SLs) enriched with medium and long chain triacylglycerols (MLCTs)	The spray dryer was operated at inlet and outlet temperatures of 180 ± 5 °C and 80 ± 5 °C, respectively with the airflow rate set at 300 NL/min	The microcapsules produced by using WPI/IN (1:1) were selected as the best treatment based on the highest microencapsulation efficiency and the lowest level of peroxide value. The finding of this study could promote the possibility of a new combination of wall materials and places IN with its high nutritional and functional properties as a substituent secondary wall material.	[42]
Gum Arabic, maltodextrin, and inulin	40–95%	Cinnamon essential oil	The inlet temperature was set at 170 °C, and the feed rate used was 0.8 L/h. An atomizing air flow rate of 35 L/min was selected and maintained.	The results demonstrated that the cinnamon essential oil microcapsules with the least amount of the surface cinnamaldehyde and with the greatest amount of the encapsulation efficiency of innam aldehyde, the release of cinnamaldehyde, and powder recovery could be efficiently produced using the selected wall materials.	[172]
Silica	70%	N-octadecane and methyl palmitate	The spray dryer was operated under a nitrogen atmosphere to prevent high-temperature combustion of the ethanol by-product of the sol-gel process 35 m^3^/h of N_2_ gas entered the dryer at 160 °C, to dehumidify the liquid droplets, and exited at 100 °C	The highest phase change materials (PCM) encapsulation efficiency could be achieved with the initial PCM to silica weight ratio of 0.25.	[59]
Arabic gum, maltodextrin, and modified starch	97.9–98.3%	Oregano essential oil	Feed rate 5 mL/min, inlet air temperature 180 °C, outlet 117 °C, aspiration 100% and airflow 600 L/h.	The release profile of essential oil (EO) from tablets prepared from spray drying powder with 10% or 20% *w*/*w* EO containing 5% *w*/*w* croscarmellose sodium was fast and similar to the profile obtained from spray-dried powder with 20% EO content.	[197]

**Table 6 pharmaceutics-15-01490-t006:** Structured lipids Encapsulation by spray drying: recent articles.

Encapsulating Agent	Encapsulation Efficiency	Structured Lipid	Encapsulation Conditions	Principal Results	Reference
Whey protein and gum Arabic	Not shown	Rosmarinus officinal	Feed flow rate of 4 g/min, atomizer 0.5 mm, inlet temperature at 90 °C, flow rate 60 m^3^/h, pressure and flow of 3 kgf/cm^2^ and 17 Lpm, respectively.	Powdered redispersible lipid-based compositions entrapping antioxidants from *Rosmarinus officinalis* extract were successfully generated by spray-drying.	[201]
Whey protein isolate (WPI) as the primary wall material by prebiotic carbohydrates, such as maltodextrin (MD) and inulin (IN)	89.10 ± 1.03%	Structured lipids (SLs) enriched with medium-and long-chain triacylglycerols (MLCTs)	Inlet and outlet temperatures of 180 °C ± 5 °C and 80 ± 5 °C, respectively with the air flow rate set at 300 NL/min.	The microcapsules produced by using WPI/IN (1:1) were selected as the best treatment based on the highest microencapsulation efficiency and the lowest level of peroxide value. The finding of this study could promote the possibility of a new combination of wall materials, and places IN with its high nutritional and functional properties as a substituent secondary wall material.	[42]
Whey protein isolate and maltodextrin	Not shown	1,3-Dioleoyl-2-palmitoylglycerol (OPO)	Inlet air temperature at 184 °C and outlet air temperature at 89 °C.	In this study, the addition of microencapsulated OPO in infant formula can significantly improve the oxidative stability of infant formula and extend the shelf life of the product. Infant formula with microencapsulated OPO possess better sense state and provide a broader space for its industrial production.	[202]
Sodium caseinate (SC), soy protein (SP) and maltodextrin (M)	42–82%	Structured lipid palm-based medium- and long-chain triacylglycerol (MLCT)	Flow rate of 15 mL/min and atomizer 0.7-mm standard diameter nozzle. The inlet and outlet temperature 140 and 110 °C, respectively.	Binary mixture of sodium caseinate and soy protein with maltodextrin when heated in solution at certain temperature and time period produces Maillard reaction products that can be used as natural emulsifier as well as encapsulating agent with improved physical properties.	[200]
Not shown	Not shown	1,3-oleic acid-2-palmitic acid structured lipid	Not shown.	The oxidative stability of the infant liquid milk added the structured lipid microcapsules has been significantly improved. Provide a theoretical basis for industrial production.	[203]
Whey protein isolate and corn syrup solids (CSS)	90%	Structured lipids (SLs) containing long-chain polyunsatured fatty acids (LCPUFAs)	Inlet temperature of 180 °C and an outlet temperature of 80 °C at a feeding rate of 5 mL/min.	Structured lipids (SLs) containing long chain polyunsatured fatty acids (LCPUFAs) were successfully microencapsulated in Maillard reaction products (MRPs) obtained from heated whey protein isolates and corn syrup solids solution with high microencapsulation efficiency.	[204]
Non-fat dry milk, whey protein isolate, lactose, maltodextrin	Not shown	Structured lipid (SL) enriched with arachidonic (ARA) and docosahexaenoic (DHA)	Two different combinations of spray-drying inlet−outlet temperature (120−70 °C vs. 180−80 °C) were used.	Formula prepared with microencapsulated SL and the dry-blending method had better oxidative stability and color quality.	[205]

## Data Availability

Not applicable.

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
