# Peer review of "Recent Advances in the Microencapsulation of Essential Oils, Lipids, and Compound Lipids through Spray Drying: A Review"

_pharmaceutics, 2023, doi:10.3390/pharmaceutics15051490_

Round 1
Reviewer 1 Report
The review article entitled "Recent advances in the microencapsulation of essential oil, lipids, and lipid compound through spray dry: a review" has an exciting theme. Still, the text is not well organized for the following reasons:
1. This manuscript must focus on addressing the main existing issues in the area of encapsulation of substances mentioned in the title; the work presents figures (quartile, e.g.) that are not so relevant to the subject in question, it would be better if it had discussed in depth the Spray Drier encapsulation techniques; The article is not in a review article structure.
2. In the last part of the introduction, the authors talk a lot about how the research was carried out. Given that the topic is broad and rich, it would have been much better if I had worked more on the content to the detriment of how the literature search was carried out.
3. Written forms of abstract, introduction and conclusion do not meet the review criteria.
(1) The abstract and the introduction should include essential parts of brief background, current problems or knowledge gaps in the related area, the purpose of the manuscript and its meaning. Current issues or knowledge gaps are lost.
(2) The conclusion should draw some views or findings from the initiative.
Overall Recommendation: reject
Author Response
"Please see the attachment."

Reviewer 2 Report
It is a well designed and well written manuscript. it can be accepted in the current form
Author Response
"Please see the attachment."

Reviewer 3 Report
Dear authors,
thanks for your submission of your review. At first look, the manuscript intends to provide a new approach to the recent advances in the encapsulation of lipids, oils and so on through the application of spray drying.
The Title must be correct, please substitute by: 'Recent advances in the microencapsulation of essential oils, lipids and compound lipids through spray drying: a review'
Abstract:
The abstract should be a total of about 200 words maximum, currently, you have 277 words - please adapt.
The abstract should be a single paragraph and should follow the style of structured abstracts but without headings: 1) Background: 2) Methods: 3) Results; and 4) Conclusion. Please remove these headings from your abstract and create a fluid paragraph(s) between all the information you already provide.
Keywords: Please change to 'Compound lipids' and 'Essential oils'
Introduction:
Lines 47-49: A paradox, could you please verify? Perhaps you want to say that 'Functional ingredients are indispensable....? Could you please verify?
Line 54: Please correct this sentence, by removing 'they';
Line 58-63: There is no suitable transition to this paragraph, please rewrite this paragraph.
Line 64: Please change to: 'Lipid, compound lipids and essential oils are...'
Line 67-69: Another paradox, please rewrite.
Line 85-86: Please remove this phrase.
Line 86: 'The main reasons are associated with i) reducing the interaction of the bioactive ingredients in different environments (heat, humidity, air and light), reducing the probability of losing bioactivity; ii) ...' - and adapt the remaining reasons.
Line 107: Please substitute 'escalation' by 'scale-up'
Line 128: '... years 2000 and 2021.'
Line 134: Change to: 'Are there significant advances in the encapsulation of essential oils, lipids or compound lipids by spray drying?'
The information between lines 124-131, should be removed from this section and should be only maintained in the Methods section. Please remove it from the introduction and rewrite the last paragraph.
Substitute throughout all manuscript 'lipid compound' by 'compound lipid'; and 'spray dry' by 'spray drying'.
Line 151: Change to 'Theoretical'
Figures: The graphic has some loss of quality, please try to add imagines with higher quality.
Figures 3 and 4 don't have the caption, please provide, as well as, the origin of the figures.
Table 1: Need to be formatted in structure, in the type of letter and the caption rewritten: 'Common encapsulating agents used in lipids (Years: 2000-2015)'
Table 2/3/4/5/6: type of letter is at 'Times New Rome', please change to MDPI typology.
Table 5, column 3, has a web connection, it is intended?
The overall manuscript is good because it shares important information on drying and formulation, but the abstract and introduction must be improved to achieve the quality of the other sections. Thus, I will recommend rewriting the indicated sections, to be considered for publication.

Author Response
"Please see the attachment."

Reviewer 4 Report
In this manuscript, the authors have compiled the literature on recent advances in the microencapsulation of essential oil, lipids, and lipid compound through spray drying.
There are a few comments as follows:
1. The introduction of the last section is needed to improve.
2. In Table 4 the encapsulation condition must be compacted.
3. Tables 2 and three must be revised
4. Table 6 must be restructured
5. Future perspective is needed before the conclusion
Author Response
"Please see the attachment."

Round 2
Reviewer 3 Report
Dear Authors,
the manuscript quality has improved, but there are some issues to solve.
Please, starting at Table 2:
- The degree symbol is changing throughout the Table, please maintain the correct format - 140 ºC (number, space, degree symbol);
- Spaces are missing throughout all tables, verify and correct;
- (32 m3.h-1), please format all of these values, sometimes you have a dot, and sometimes you have a superscript, and sometimes you have '/'. Thus, correct everything using the same rules.
- 40% or 40 %, use the same rule throughout the manuscript.
Please verify all Tables and the manuscript in these format questions.
Also, throughout the manuscript there are several abbreviations, perhaps would be helpful to do an abbreviation list, please follow MDPI's written rules.
Author Response
"Please see the attachment."

Reviewer 4 Report
The revised manuscript can be proceeded for further process in journal.
Author Response
"Please see the attachment."
